# The Economic Impact of Post-Traumatic Stress Disorder Among Patients with Systemic Autoimmune Diseases During the COVID-19 Pandemic

**DOI:** 10.3390/ijerph21111476

**Published:** 2024-11-06

**Authors:** Valentina Lorenzoni, Gianni Andreozzi, Ilaria Palla, Chiara Tani, Claudia Carmassi, Giovanni Fulvio, Francesca Trentin, Sara Fantasia, Lorenzo Conti, Liliana Dell’Osso, Marta Mosca, Giuseppe Turchetti

**Affiliations:** 1Intitute of Management, Scuola Superiore Sant’Anna, 56127 Pisa, Italy; gianni.andreozzi@santannapisa.it (G.A.); ilaria.palla@santannapisa.it (I.P.); giuseppe.turchetti@santannapisa.it (G.T.); 2Rheumatology Unit, Department of Clinical and Experimental Medicine, University of Pisa, 56126 Pisa, Italy; chiaratani78@gmail.com (C.T.); giovanni.fulvio92@gmail.com (G.F.); francesca3ntin@gmail.com (F.T.); marta.mosca@unipi.it (M.M.); 3Psychiatry Unit, Department of Clinical and Experimental Medicine, University of Pisa, 56126 Pisa, Italy; claudia.carmassi@unipi.it (C.C.); dr.fantasiasara@gmail.com (S.F.); lorenzo.conti@unipi.it (L.C.); liliana.dellosso@med.unipi.it (L.D.)

**Keywords:** COVID-19, post-traumatic stress disorder, systemic autoimmune disease, economic, healthcare costs, indirect costs, LASSO regression

## Abstract

Background: The present work aimed to estimate the economic impact of PTSD following COVID-19 in a population of patients affected by systemic autoimmune disease (SAD) using a cost-of-illness approach and accounting for the perspective of society. Methods: Considering data collected from SAD patients enrolled in a specialized outpatient clinic in the Tuscany region, Italy, generalized linear models and LASSO logistic regression were used to evaluate the impact of PTSD on costs and its relevance as a possible predictor of being a high-cost patient, respectively. Results: Considering 301 SAD patients, 161 (51.2%) of whom were diagnosed with PTSD, the overall costs were EUR 3670 [890; 40,529] per patient/year among patients with PTSD and EUR 2736.7 [283; 21,078] per patient/year among those without PTSD (*p*-value < 0.001), with differences mainly attributable to significantly greater direct non-healthcare and indirect costs. PTSD was estimated to increase overall costs (β = 0.296 (0.140), *p*-value = 0.035), direct non-healthcare (β = 1.193 (0.392), *p*-value = 0.002), and indirect costs (β = 3.741 (1.136), *p*-value = 0.001). PTSD diagnosis was also significantly associated with the likelihood of being a high-cost patient. Conclusions: Findings from the present study offer a novel perspective on the economic impact of COVID-19 and provide valuable data for policymakers to better understand the demand for healthcare services and associated costs.

## 1. Introduction

The COVID-19 pandemic was an unprecedented event that has affected the entire world and exposed the entire Western region to fears never known before in current generations, undermining much of the acquired security and social life and compromising the capacity of health services [1]. Indeed, in addition to the health consequences of COVID-19 infections, restrictive measures set in many countries have exacerbated the impact of COVID-19 by further disrupting habits and social life [2].

The result has been an impact on the psychophysical condition of the population, particularly fragile subgroups [3]. High levels of distress, sleep disturbances, anxiety, depressive or obsessive symptoms, and post-traumatic stress disorder (PTSD) have been reported in several studies [4,5,6,7,8,9].

In particular, studies addressing the initial phase of COVID-19 have reported increased symptoms of PTSD, with a greater risk of developing PTSD in certain subgroups [10], including people affected by systemic autoimmune disease (SAD) [11,12,13,14]. Moreover, the effects of COVID-19 on mental health were also confirmed in subsequent phases. There is indeed a large consensus that COVID-19 outbreaks have a wide range of psychological impacts, including PTSD in individuals not suffering from mental illnesses and exacerbation of psychiatric symptoms in those with preexisting mental illnesses [15] and SAD patients [16].

The economic consequences of PTSD following COVID-19 in the general population and in patients with SAD are understudied at present, even though effects on the healthcare system and society will indeed span years [17].

In particular, not only will healthcare systems and society be affected by the economic burden of mental illness induced by the pandemic, but those consequences are likely to contribute to socioeconomic crises [18,19].

Therefore, the present work aimed to provide estimates of the economic impact of PTSD following COVID-19 in a population of patients affected by SAD and treated in a specialized outpatient clinic in the Tuscany region, Italy. The specific objectives of this paper are to estimate the economic impact of PTSD by accounting for potential associated factors and considering a cost-of-illness approach [20,21]; to evaluate the impact of PTSD on different cost components, thus valuing direct healthcare and non-healthcare costs as well as indirect costs from productivity loss, by means of generalized linear models (GLMs) to also account for the effects of other dependent variables [22]; to assess the relevance of PTSD as a possible predictor of being a high-cost patient considering all the different cost components; and to use least absolute shrinkage and selection operator (LASSO) logistic regression for the selection of relevant predictors.

## 2. Materials and Methods

### 2.1. Design and Setting

This empirical study is conducted in the context of a project funded by the Tuscany region in Italy, the PERMAS project. The PERMAS project is an observational study focused on SAD patients, the general aims of which are to evaluate the clinical psychopathological and socioeconomic impacts of emergency situations on SAD patients and to design appropriate intervention strategies. Adult subjects who were diagnosed with SAD and followed up at the Rheumatology Unit of the University of Pisa were enrolled in the study at the time of their planned outpatient visit. At the time of enrollment, clinical and socioeconomic data related to the COVID-19 pandemic were collected. The study was conducted in accordance with the directives issued by the Declaration of Helsinki and the Nuremberg Code and approved by the Regional Ethical Committee for Experimentation Clinic of the Tuscany Region, Area Vasta Nord Ovest section on 11 February 2021, protocol No. 19233.

The study sample considered in the present evaluation included 301 SAD patients who were consecutively enrolled from May 2021 to July 2022. All patients were enrolled in the study after providing written informed consent.

The exclusion criteria were any limitation in verbal communication that compromised the subject’s ability to follow the protocol assessment (for example, poor knowledge of the Italian language) and/or lack of collaboration skills.

### 2.2. Data Collection

At enrollment, patients were asked to complete a dedicated questionnaire including ad hoc questions as well as validated instruments to collect information about sociodemographic data, COVID-19 pandemic-related information, a psychopathological evaluation to assess the impact of the pandemic, facts related to the use of healthcare services, and the need for assistance in the six months preceding the assessment.

The Trauma and Loss Spectrum-Self Report (TALS-SR) was used to examine post-traumatic stress symptoms.

In detail, the Italian version of the TALS-SR [23,24] investigates the typical, atypical, attenuated, and subthreshold symptoms related to the post-traumatic stress spectrum, providing a complete dimensional approach to the psychophysiology of the individual. This self-assessment questionnaire includes 116 items with a dichotomous response (yes/no) organized into nine domains: (1) loss events, (2) grief reactions, (3) potentially traumatic events, (4) reactions to losses or upsetting events, (5) re-experiencing, (6) avoidance and numbing, (7) maladaptive coping, (8) arousal, and (9) personal characteristics/risk factors. The TALS-SR was adapted for detecting post-traumatic stress spectrum symptoms related to the COVID-19 pandemic. In line with previous studies [25,26,27], symptomatologic PTSD incidence rates according to the DSM-5 criteria were assessed by means of matching between the TALS-SR and the DSM PTSD symptoms.

According to previous literature on DSM-5 criteria, a partial PTSD diagnosis was also considered by means of the fulfillment of three out of four of the B-E symptomatologic criteria [28]. The TALS-SR presented good intraclass correlation coefficients (from 0.934 to 0.994) with the SCI-TALS, the interview version adopted for assessing post-traumatic stress symptomatology. Similarly, the SCI-TALS showed good internal consistency (a Kuder–Richardson coefficient exceeding the minimum standard of 0.50 for each domain).

An electronic version of the questionnaire was implemented using EUsurvey, and dedicated workstations with tablets were arranged at the enrolling center to allow patients to complete self-assessments while waiting for their usual follow-up visit.

### 2.3. Costs Valuation

The questionnaire included specific questions aimed at assessing resource use related to direct and indirect costs over the 6 months preceding the date of assessment.

In detail, data on resource use related to hospital admissions, emergency department (ED) visits, outpatient care (specialist visits, laboratory exams, and imaging evaluations), drugs and out-of-pocket expenses, including patients’ additional payments for prescribed treatments, and other disease-related expenses were collected to derive direct healthcare costs. Moreover, hours of formal and informal assistance, costs for travel and accommodation because of BS-related consultations, and working days lost were collected to estimate direct non-healthcare costs and indirect costs for productivity loss.

Cost assessment was performed from the perspective of society.

Direct healthcare cost resources related to hospital admission, ED visits, and outpatient care were valued according to national price lists [29]. The costs related to prescribed drugs were estimated considering drug dosage, frequency of use collected from the questionnaire, and national list prices [30].

Direct non-healthcare costs pertaining to transportation to healthcare providers and lodging expenses were listed by the patients in the questionnaire.

Costs for formal and informal assistance were estimated using the proxy good method, which values the care provided by the informal caregiver, considering that if he/she did not provide these services, his/her presence would have to be substituted by another person who could provide them, thus considering the hourly wage of the formal caregiver, as reported by Eurostat [31], and the hours of assistance needed, as listed by patients in the questionnaire.

Indirect costs were made up of productivity losses due to job loss or absenteeism and estimated using the human capital approach, considering working days lost declared by patients and hourly wages from annual net earnings for Italy from Eurostat [31].

Costs over a 1-year period were estimated as twice the cost estimated for 6 months from data collected from the questionnaire. All costs were referred to Euro 2023.

### 2.4. Statistical Analysis

The main characteristics of the study sample are described as the mean and standard deviation for quantitative measures and the number of subjects and percentage for categorical variables.

The costs are presented as means, minimum and maximum values, and medians along with the 25th and 75th percentiles, given their skewed distributions.

Comparisons between groups were performed using the chi-squared test, independent samples Student’s *t* test, and Mann-Whitney test, as appropriate.

The impact of PTSD on the different cost components was evaluated using GLMs.

GLMs are a broad class of models that relax the ordinary least squares (OLS) assumption used in traditional linear models to allow robust and consistent modeling of response variables with heteroscedastic and/or skewed distributions [32], as is generally the case for cost data.

In detail, GLMs consider the random part of the model following an exponential distribution (i.e., Gaussian, gamma, Weibull, and inverse Gaussian) and make use of an invertible link function *g*(.) to relate the expectation of the response variable *E*(*y_i_*) to a linear combination of the covariates:$$g\left(E(y_{i})\right)=x_{i}\beta$$

Parameter estimation was obtained using maximum likelihood estimation (MLE) [32].

The use of a gamma distribution in GLM is particularly useful for modeling right-skewed responses constrained in the positive interval and has been shown to perform well with cost data [22,33].

The gamma probability distribution function is characterized by the shape *a* and scale *b* parameters:$$f\left(Y\right)=\frac{1}{b^{a}\Gamma (a)}\;y^{\left(a-1\right)\;}e^{-(\frac{y}{b})}$$
and the mean and the variance are functions of both scale parameters, as follows:$$E\left(Y\right)=a*b$$
$$Var\left(Y\right)=a*b^{2}$$

Therefore, the ratio of the variance to the mean is constant.

Accordingly, considering that different response variables $y_{i}$ are overall costs or one of the different cost components evaluated in the present analysis for subject *i*, a gamma distribution and log link function were used to relate $E\left(y_{i}\right)\;$ to the linear combination of a set of *j* predictors $x_{ij}$ evaluated for each subject i and included the intercept:$$\left(E(y_{i}|x_{i})\right)={exp(x}_{i}\beta )$$

The predictors selected in the present analysis comprised the main variable of interest, PTSD diagnosis, and a series of additional predictors related to sociodemographic, clinical, and COVID-19-related data, which are potential confounders; see details in Table 1.

For all models, the pseudo R^2^ was obtained to evaluate model fitting, and collinearity was evaluated by the variance inflation factor (VIF).

To evaluate the relevance of PTSD as a predictor of high-cost patients, LASSO logistic regression [34] was used. Indeed, the LASSO method is widely used for high-dimensional predictor selection, and it has been shown to outperform conventional methods of variable selection, even in the presence of limited data.

Considering a response variable $y_{i}$ observed over *i* subjects and a set of a set of *j* predictors $x_{ij}$ evaluated for each subject *i*, the LASSO solves a l^1^-penalized regression problem of finding coefficients for each of the j predictors, minimizing the following:$$\sum_{i}{\left(y_{i}-\sum_{j}x_{ij}\beta_{j}\right)}^{2}+\lambda \sum_{j}\left|\beta_{j}\right|$$

In other words, in contrast to OLS regression, LASSO regression introduces a penalty to perform variable selection and shrinkage.

The response variables used in the LASSO logistic regression and identifying high-cost patients were dummy variables obtained for overall costs and for the different cost components, taking a value of 0 or 1 for those for which costs were lower or equal to or greater than a certain percentile of the distribution, respectively.

Given the size of the study sample and to assess the consistency of the findings, the 75th, 90th, and 95th percentiles of the distributions of the different cost components were considered to create dummy variables to be used in LASSO logistic regressions. The same predictors used in the GLMs were considered in the LASSO logistic regression: the main variable of interest, PTSD diagnosis, and a series of additional predictors related to sociodemographic, clinical, and COVID-19-related data were considered potential confounders (Table 1).

The study sample was divided into training and test groups (comprising 70% and 30%, respectively, of the overall sample), and the model showing the minimum lambda in the training group was identified through cross-validation and then used in the test group.

All analyses were conducted using R software version 4.2.3, and the glmnet package was used for LASSO logistic regression.

## 3. Results

A total of 301 SAD patients were considered in the present study, among whom 161 (51.2%) had full-blown or partial PTSD, as evaluated by the TALS-SR.

SAD patients with full-blown or partial PTSD were younger (*p*-value = 0.002), more likely to be female (*p*-value < 0.001), and more likely to be housekeepers or students (*p*-value < 0.001) who were mainly diagnosed with connective tissue disease (CTD) (*p*-value = 0.001) and prescribed psychiatric treatment (*p*-value = 0.020).

A detailed description of the study sample according to PTSD status is reported in Table 2.

Overall costs were EUR 3670 [890; 40,529] per patient/year among SAD patients with PTSD and EUR 2736.7 [283; 21,078] per patient/year among those without PTSD (*p*-value < 0.001).

In terms of the different cost components, significant differences also emerged between the groups. In detail, SAD patients who also had PTSD had significantly greater direct non-healthcare costs (EUR 1108 [0; 20,065] per patient/year vs. EUR 523 [0; 16,620] per patient/year, *p*-value = 0.001) and indirect costs (EUR 380 [0; 17,265] per patient/year vs. EUR 27.9 [0; 1381] per patient/year, *p*-value < 0.001), while no differences were found for direct healthcare costs (EUR 2183 [214; 6798] per patient/year vs. EUR 2186 [44; 4461] per patient/year) (Figure 1).

The figure shows a breakdown of the mean overall costs per patient/year on the basis of the different costs associated with PTSD diagnosis. The numbers detail the mean costs according to the specific cost component and the relative contribution to the overall mean cost per patient/year.

Accordingly, while among SAD patients not showing PTSD, direct healthcare costs contributed largely to the overall mean cost per patient/year (79.9%), among those exhibiting PTSD, direct healthcare costs represented 59.5% of the overall mean costs per patient/year, and direct non-healthcare costs and indirect costs accounted for 30.2% and 10.3% of the overall costs, respectively.

Analysis of the impact of PTSD by GLM and adjustment for potential confounders revealed that PTSD increased overall costs (β = 0.296 (0.140), *p* = 0.035), direct non-healthcare costs (β = 1.193 (0.392), *p* = 0.002), and indirect costs (β = 3.741 (1.136), *p* = 0.001), while PTSD did not affect direct healthcare costs (β = 0.031 (0.041), *p* = 0.459) (Table 3, Table 4, Table 5 and Table 6).

The different factors selected by the LASSO logistic regression analysis for the different percentiles considered and the diverse cost components are shown in Table 7, Table 8, Table 9 and Table 10.

Consistent with other analyses, PTSD did not emerge as a predictor for being a high-cost patient when focusing on high direct healthcare costs, while it emerged as a predictor for overall costs, direct non-healthcare costs, and indirect costs when considering all the different percentiles, and there was a greater likelihood of those diagnosed with PTSD being high-cost patients.

In particular, for overall costs, the coefficients associated with PTSD were 0.666, 0.682, and 0.097 for the 75th, 90th, and 95th percentiles of the distribution, respectively (Table 7).

For direct non-healthcare costs, the coefficients associated with PTSD were 0.662, 1.252, and 0.065 for the 75th, 90th, and 95th percentiles of the distribution, respectively (Table 9).

Finally, for indirect costs, the coefficients related to PTSD were 0.519, 0.090, and 1.746 for the 75th, 90th, and 95th percentiles of the distribution, respectively (Table 10).

## 4. Discussion

The present study aimed to use a cost-of-illness approach to evaluate the economic impact of pandemic-related PTSD in patients with SAD to understand its effects on specific cost components and the likelihood of being a high-cost patient and to provide valuable data for policymakers to better target interventions and understand their value.

Indeed, PTSD has been shown to impose a significant burden on patients, their caregivers, and society because of the associated physical impairment, tendency toward a chronic course with often limited response to available treatments, comorbid psychiatric disorders, and relevant impacts on work and social functioning [35,36]. However, literature on the full economic dimension of PTSD is scarce, particularly in Europe and in samples different from those of veterans and elderly individuals [37].

The findings from the present study highlight that among SAD patients, pandemic-related PTSD implies an excess economic burden of the disease for society.

The COVID-19 pandemic significantly shaped life and habits worldwide and impacted the health and psychological well-being of people, with consequences lasting long after the acute phase of the pandemic.

The pandemic not only disrupted economic activity, resulting in market volatility, especially in developed countries [38], but also affected economic and healthcare costs, and there is a need to increase knowledge about all possible economic implications and to adequately invest in preventive and preparedness measures to mitigate the risks of large epidemics.

Indeed, despite the broad impact on the main economic indicators [39], the long-term effects of the pandemic will have an impact on public health and the economy for years to come, not least in view of the consequences of COVID-19 on the health and mental health of the general population and, in particular, vulnerable subgroups such as those with SAD.

The socioeconomic literature on this topic is currently limited, and SAD has frequently been overlooked [40].

The present study aims to fill this gap by providing valuable information for policymakers.

In detail, in addition to higher costs among SAD patients exhibiting PTSD, findings from the present study suggest that the impact of PTSD was mainly on direct non-health and indirect costs; the impact was observed both for the magnitude of costs and for the likelihood of being a high-cost patient.

Although those observations seem to contrast with the available literature on the economic dimension of PTSD, which generally found an excess of healthcare costs associated with PTSD [17,37,41,42], the effect found in the present study needs to be interpreted considering different possible explanations pertaining to both the COVID-19 pandemic and the general behavior related to psychological and psychiatric treatment. First, during the different waves of the pandemic, the provision of care for chronic illnesses was disrupted, especially for SAD patients, who were disproportionately affected by a reduction in the continuity of care because of the redeployment of rheumatology staff, concerns about the availability of treatment, and the general behavior of avoiding seeking care because of the fear of infection and related consequences [43]. Of note, among the few studies providing an evaluation of the economic impact on PTSD comprising costs other than those related to healthcare-resource use, Ferry et al. (2005) [41] showed that productivity losses and presenteeism account for more than 80% of the overall costs of PTSD in the general population.

In addition, previous findings reported increasing direct healthcare costs among PTSD patients, mainly because of psychiatric treatment, and it is important to consider that, depending on the context and the organization and availability of mental health services, because of social stigma, psychological and psychiatric services are sometimes not sought [43]; this may also be in line with the low proportion of patients in our sample receiving psychiatric therapy (6.3% overall and 9.4% among those with full-blown or partial PTSD). To further elaborate on the fact that PTSD does not have a significant impact on healthcare costs in our study, additional explanations can also be adduced. In particular, given the generally high direct healthcare costs of SAD patients, PTSD may have a relatively low impact on these costs; another possible explanation could be related to the relatively short period of time considered after PTSD diagnosis. On the other hand, findings from the present study are in line with both previous studies suggesting a high impact of PTSD on individuals with significant impairment in global social functioning [44,45,46,47] and evidence about the large contribution of costs other than direct healthcare costs to the overall economic burden of PTSD [41].

Notably, similar observations have been confirmed in SAD patients, and these patients deserve special attention because they are highly vulnerable to functional and social impairments because of the effects of their disease. Indeed, PTSD is likely to exacerbate the vulnerability of SAD patients, thus aggravating the condition in certain people, mainly women and those who are already limited in their daily and social activities, while increasing the economic burden of the disease.

Although the present work focused on PTSD, with respect to the impact on costs, changes in psychiatric therapy during COVID-19 were also associated with higher overall costs and indirect costs as well as the probability of being a high-cost patient, as these findings corroborate the economic impact of psychiatric symptoms following COVID-19 [48].

Some limitations of the present study need to be acknowledged.

First, our empirical evaluation relies on an observational study in which the collection of data was on a voluntary basis, limiting the representativeness of the study sample because of a possible selection bias caused by the willingness to participate in the study; moreover, retrospective collection of data may be subject to recall bias.

The observational nature of the study and the design, which did not include prepandemic data, limit the comparability of the two groups; few differences were found with respect to baseline characteristics, and multivariate models were used to control for possible confounders.

Finally, the size of the study sample limits some analysis despite the authors trying to address this issue with appropriate statistical methods.

Despite the limitations, the present paper expands the current literature on the economic impact of PTSD with the specific purpose of focusing on pandemic-related PTSD among SAD patients, a subgroup that, per se, is more at risk of developing PTSD due to the specific condition from which they suffer and because they are predominantly women.

A major innovation of our work is the evaluation of the economic impact of PTSD using a cost-of-illness approach and accounting for the perspective of society, accounting for costs other than direct healthcare costs, and the use of proper statistical methods to evaluate the effect of PTSD on costs, accounting for the effects of other potential variables.

In summary, we build on the previous literature and improve on it in different ways. First, we focused on a different vulnerable group—at present overlooked—being inclusive with respect to how the different cost components of PTSD may affect and be relevant to society. Moreover, we evaluated the impact of PTSD in SAD patients on the different cost components, also accounting for the potential effects of other relevant variables, and we also assessed the potential effect of PTSD in predicting high-cost patients, providing relevant information for policymakers.

## 5. Conclusions

To conclude, in a sample of SAD patients, our empirical evaluation highlights the impact of PTSD on the overall costs of the disease and on specific cost components pertaining to direct non-healthcare costs and indirect costs. Similarly, PTSD was found to be a relevant predictor of being a high-cost patient considering the same cost components.

Given the impact of COVID-19 on mental health in specific subgroups, there is a need to increase awareness among policymakers, planners, commissioners, service providers, service users, and the wider public of the extent of the public health and economic burden of PTSD in different population groups and the associated adverse economic implications to better target policies and correctly frame health economy evaluations of possible interventions targeting PTSD.

In subgroups such as SAD, it is important that all specialists involved in the treatment of the disease consider possible psychiatric symptoms related to the impact of the pandemic to direct patients to appropriate treatment to manage their psychiatric problems, which will have a favorable impact on costs for both the national health system and society.

The present work may also offer insights into the management of future shocks in similar patients but also serves to reinforce the existing literature on the impact of PTSD following shocks in vulnerable populations in general.

## Figures and Tables

**Figure 1 ijerph-21-01476-f001:**
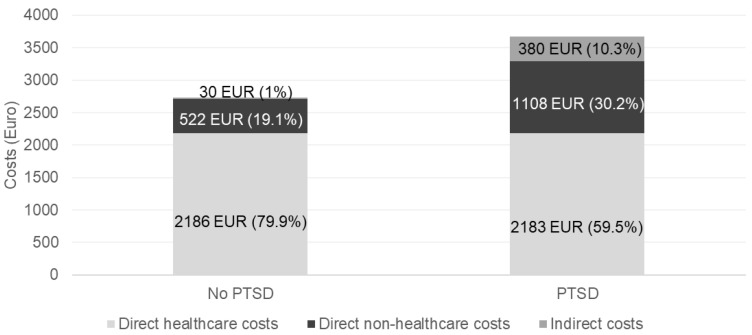
Costs per patient/year according to the different costs associated with PTSD diagnosis.

**Table 1 ijerph-21-01476-t001:** Independent variable used in the model.

Variable	Type of Variable	Value Label
Sociodemographic variables		
Age	Continuous	Attained age in years
Gender	Dummy	Male
		Female
Marital status	Categorical	Single
		Married/Cohabitant
		Divorced/Widowed
Working status	Categorical	Employed
		Unemployed
		Retired
		Other (housekeeper, student)
Clinical variable		
Rheumatic diagnosis	Categorical	CTD
		Inflammatory arthritis
		Systemic vasculitis
Ongoing psychiatric therapy	Dummy	No
		Yes
COVID-19 related variables		
Being infected with COVID-19	Dummy	No
		Yes
Immunosuppressive treatment suspended during COVID-19 pandemic	Dummy	No
		Yes
Psychiatric therapy modified during COVID-19 pandemic	Dummy	No
		Yes

CTD = connective tissue disease.

**Table 2 ijerph-21-01476-t002:** Main characteristics of the study sample according to PTSD diagnosis.

	No PTSD Diagnosis (N = 141)	PTSD Diagnosis (N = 160)	Overall (N = 301)
Age ^1^	54.6 (13.2)	49.9 (12.8)	52.1 (13.2)
Gender ^1^			
Male	53 (37.6%)	23 (14.4%)	76 (2.3%)
Female	88 (62.4%)	137 (85.6%)	225 (74.7%)
Experiencing COVID-19			
No	117 (83%)	144 (90%)	261 (86.7%)
Yes	24 (17%)	16 (10%)	40 (13.3%)
Rheumatic diagnosis ^1^			
CTD	56 (39.7%)	99 (61.9%)	155 (51.5%)
Inflammatory arthritis	71 (50.4%)	51 (31.9%)	122 (40.5%)
Systemic vasculitis	14 (9.9%)	10 (6.3%)	24 (8%)
Marital status			
Single	21 (14.9%)	34 (21.3%)	55 (18.3%)
Married/Cohabitant	101 (71.6%)	106 (66.3%)	207 (68.8%)
Divorced/Widowed	19 (13.5%)	20 (12.5%)	39 (12.9%)
Educational level			
Low	41 (29.1%)	45 (28.1%)	86 (28.6%)
Middle	67 (47.5%)	82 (51.3%)	149 (49.5%)
High	33 (23.4%)	33 (20.6%)	66 (21.9%)
Working status ^1^			
Employed	80 (56.7%)	89 (55.6%)	169 (56.2%)
Unemployed	8 (5.7%)	14 (8.8%)	22 (7.3%)
Retired	41 (29.1%)	15 (9.4%)	56 (18.6%)
Other (housekeeper, student)	12 (8.5%)	42 (26.3%)	54 (17.9%)
Ongoing psychiatric therapy ^1^			
No	137 (97.2%)	145 (90.6%)	282 (93.7%)
Yes	4 (2.8%)	15 (9.4%)	19 (6.3%)
Psychiatric therapy modified during COVID-19 pandemic			
No	139 (98.6%)	153 (95.6%)	292 (97%)
Yes	2 (1.45)	7 (4.4%)	9 (3%)
Immunosuppressive treatment suspended during COVID-19 pandemic			
No	104 (73.8%)	124 (77.5%)	228 (75.8%)
Yes	37 (26.2%)	36 (22.5%)	73 (24.2%)

^1^*p*-value < 0.05. CTD = connective tissue disease; PTSD = post-traumatic stress disorder.

**Table 3 ijerph-21-01476-t003:** Overall costs. Results from the generalized linear model.

	Overall Costs		
	Coef. (Std. Err.)	95% CI	*p*-Value
PTSD diagnosis			
No	*(ref)*		
Yes	**0.296 (0.140)**	**(−0.003; 0.543)**	**0.035**
Age	−0.009 (0.006)	(−0.021; 0.004)	0.169
Gender			
Female	*(ref)*		
Male	−0.213 (0.171)	(−0.548; 0.121)	0.211
Marital status			
Single	*(ref)*		
Married/Cohabitant	0.041 (0.201)	(−0.353; 0.434)	0.840
Divorced/Widowed	−0.064 (0.268)	(−0.59; 0.461)	0.810
Working status			
Employed	*(ref)*		
Unemployed	−0.322 (0.263)	(−0.837; 0.194)	0.221
Retired	0.38 (0.208)	(−0.027; 0.787)	0.067
Other	−0.223 (0.186)	(−0.588; 0.142)	0.232
Rheumatic diagnosis			
CTD	*(ref)*		
Inflammatory arthritis	0.129 (0.155)	(−0.174; 0.432)	0.405
Systemic vasculitis	0.153 (0.257)	(−0.351; 0.657)	0.552
Ongoing psychiatric therapy			
No	*(ref)*		
Yes	−0.457 (0.361)	(−1.165; 0.251)	0.206
Being infected with COVID-19			
No			
Yes	−0.193 (0.206)	(−0.598; 0.211)	0.349
Psychiatric therapy modified during COVID-19 pandemic			
No	*(ref)*		
Yes	**1.284 (0.524)**	**(0.257; 2.311)**	**0.014**
Immunosuppressive treatment suspended during COVID-19 pandemic			
No	*(ref)*		
Yes	0.272 (0.169)	(−0.059; 0.603)	0.107
Constant	**8.259 (0.314)**	**(7.644; 8.874)**	**0.000**

CTD = connective tissue disease; PTSD = post-traumatic stress disorder. Bold format: Statistically significant data are in bold.

**Table 4 ijerph-21-01476-t004:** Direct healthcare costs. Results from the generalized linear model.

	Direct Healthcare Costs		
	Coef. (Std. Err.)	95% CI	*p*-Value
PTSD diagnosis			
No	*(ref)*		
Yes	0.031 (0.041)	(−0.051; 0.112)	0.459
Age	−0.003 (0.002)	(−0.007; 0.001)	0.081
Gender			
Female	*(ref)*		
Male	0.043 (0.050)	(−0.056; 0.142)	0.397
Marital status			
Single	*(ref)*		
Married/Cohabitant	−0.004 (0.058)	(−0.118; 0.109)	0.940
Divorced/Widowed	−0.064 (0.076)	(−0.214; 0.085)	0.399
Working status			
Employed	*(ref)*		
Unemployed	−0.122 (0.077)	(−0.273; 0.029)	0.114
Retired	0.017 (0.064)	(−0.108; 0.141)	0.794
Other	−0.040 (0.055)	(−0.146; 0.067)	0.468
Rheumatic diagnosis			
CTD	*(ref)*		
Inflammatory arthritis	0.127 (0.044)	(0.041; 0.213)	0.004
Systemic vasculitis	0.193 (0.077)	(0.042; 0.344)	0.012
Ongoing psychiatric therapy			
No	*(ref)*		
Yes	−0.134 (0.108)	(−0.345; 0.077)	0.214
Being infected with COVID-19			
No	*(ref)*		
Yes	−0.074 (0.059)	(−0.19; 0.041)	0.207
Psychiatric therapy modified during COVID-19 pandemic			
No	*(ref)*		
Yes	0.232 (0.155)	(−0.071; 0.535)	0.133
Immunosuppressive treatment suspended during COVID-19 pandemic			
No	*(ref)*		
Yes	0.087 (0.048)	(−0.007; 0.181)	0.069
Constant	7.784 (0.097)	(7.594; 7.974)	<0.001

CTD = connective tissue disease; PTSD = post-traumatic stress disorder.

**Table 5 ijerph-21-01476-t005:** Direct non-healthcare costs. Results from the generalized linear model.

	Direct Non-Healthcare Costs		
	Coef. (Std. Err.)	95% CI	*p*-Value
PTSD diagnosis			
No	*(ref)*		
Yes	1.193 (0.392)	(0.425; 1.961)	0.002
Age	−0.032 (0.017)	(−0.066; 0.002)	0.063
Gender			
Female	*(ref)*		
Male	−2.524 (0.507)	(−3.517; −1.531)	<0.001
Marital status			
Single	*(ref)*		
Married/Cohabitant	0.444 (0.586)	(−0.704; 1.592)	0.448
Divorced/Widowed	0.783 (0.788)	(−0.762; 2.328)	0.321
Working status			
Employed	*(ref)*		
Unemployed	−1.337 (0.737)	(−2.781; 0.107)	0.070
Retired	2.226 (0.582)	(1.085; 3.367)	<0.001
Other	−1.283 (0.539)	(−2.338; −0.227)	0.017
Rheumatic diagnosis			
CTD	*(ref)*		
Inflammatory arthritis	0.027 (0.506)	(−0.965; 1.019)	0.957
Systemic vasculitis	1.157 (0.725)	(−0.264; 2.578)	0.110
Ongoing psychiatric therapy			
No	*(ref)*		
Yes	−1.674 (1.005)	(−3.643; 0.295)	0.096
Being infected with COVID-19			
No	*(ref)*		
Yes	−1.434 (0.659)	(−2.725; −0.142)	0.030
Psychiatric therapy modified during COVID-19 pandemic			
No	*(ref)*		
Yes	3.581 (1.481)	(0.678; 6.484)	0.016
Immunosuppressive treatment suspended during COVID-19 pandemic			
No	*(ref)*		
Yes	1.030 (0.498)	(0.053; 2.006)	0.039
Constant	7.166 (0.782)	(5.633; 8.700)	<0.001

CTD = connective tissue disease; PTSD = post-traumatic stress disorder.

**Table 6 ijerph-21-01476-t006:** Indirect costs. Results from the generalized linear model.

	Indirect Costs		
	Coef. (Std. Err.)	95% CI	*p*-Value
PTSD diagnosis			
No	*(ref)*		
Yes	3.741 (1.136)	(1.514; 5.967)	0.001
Age	−0.071 (0.054)	(−0.177; 0.035)	0.189
Gender			
Female	*(ref)*		
Male	2.442 (1.651)	(−0.794; 5.677)	0.139
Marital status			
Single	*(ref)*		
Married/Cohabitant vs. Single	2.006 (1.311)	(−0.564; 4.576)	0.126
Divorced/Widowed vs. Single	2.252 (1.718)	(−1.116; 5.619)	0.190
Working status			
Employed	*(ref)*		
Unemployed	−3.042 (3.204)	(−9.322; 3.239)	0.342
Retired	−6.523 (3.099)	(−12.598; −0.449)	0.035
Other	−5.299 (1.427)	(−8.096; −2.503)	<0.001
Rheumatic diagnosis			
CTD	*(ref)*		
Inflammatory arthritis	2.790 (1.913)	(−0.959; 6.538)	0.145
Systemic vasculitis	0.601 (3.137)	(−5.547; 6.749)	0.848
Ongoing psychiatric therapy			
No	*(ref)*		
Yes	−3.046 (2.260)	(−7.476; 1.384)	0.178
Being infected with COVID-19			
No	*(ref)*		
Yes	−3.141 (1.56)	(−6.199; −0.083)	0.044
Psychiatric therapy modified during COVID-19 pandemic			
No	*(ref)*		
Yes	10.655 (3.171)	(4.44; 16.869)	0.001
Immunosuppressive treatment suspended during COVID-19 pandemic			
No	*(ref)*		
Yes	0.732 (1.654)	(−2.510; 3.974)	0.658
Constant	4.506 (2.502)	(−0.398; 9.410)	0.072

CTD = connective tissue disease; PTSD = post-traumatic stress disorder.

**Table 7 ijerph-21-01476-t007:** Patients with high overall costs. Factors selected by the LASSO logistic regression model.

	75th Percentile	90th Percentile	95th Percentile
Factors	Coefficient	Coefficient	Coefficient
PTSD diagnosis	0.666	0.682	0.097
Male	-	−0.010	-
Inflammatory arthritis	-	−0.268	
Systemic vasculitis	−0.135	0.697	-
Divorced/Widowed	-	−1.045	-
Psychiatric therapy modified during COVID-19 pandemic	1.002	1.224	2.128
Being infected with COVID-19	-	−0.484	-

PTSD = post-traumatic stress disorder.

**Table 8 ijerph-21-01476-t008:** Patients with high direct healthcare costs. Factors selected by the LASSO logistic regression model.

	75th Percentile	90th Percentile	95th Percentile
Factors	Coefficient	Coefficient	Coefficient
Age	-	−0.001	−0.025
CTD	−0.334	-	2.006
Systemic vasculitis	1.706	2.117	2.138
Unemployed	-	-	−0.134
Retired	−0.445	-	-
Other working status	-	-	0.063
Married/Cohabitant	0.382	-	1.261
Divorced/Widowed	−0.285	-	-
Ongoing psychiatric therapy	0.198	-	-
Immunosuppressive treatment suspended during COVID-19 pandemic	0.296	-	-

CTD = connective tissue disease.

**Table 9 ijerph-21-01476-t009:** Patients with high direct non-healthcare costs. Factors selected by the LASSO logistic regression model.

	75th Percentile	90th Percentile	95th Percentile
Factors	Coefficient	Coefficient	Coefficient
PTSD diagnosis	0.662	1.252	0.065
Male	−0.250	-	-
Inflammatory arthritis	−1.033	−0.172	-
Systemic vasculitis	-	0.203	-
Retired	0.229	0.736	-
Other working status	−0.452	-	-
Single	0.253	-	-
Married/Cohabitant	-	-	-
Divorced/Widowed	−0.385	−1.430	-
Being infected with COVID-19	0.607	−0.752	-
Psychiatric therapy modified during COVID-19 pandemic	0.408	1.253	0.832

PTSD = post-traumatic stress disorder.

**Table 10 ijerph-21-01476-t010:** Patients with high indirect costs. Factors selected by the LASSO logistic regression model.

	75th Percentile	90th Percentile	95th Percentile
Factors	Coefficient	Coefficient	Coefficient
PTSD diagnosis	0.519	0.090	1.746
Male	−0.012	-	-
Systemic vasculitis	0.652	-	−0.411
Employed	0.821	0.283	-
Other working status	-	-	−0.711
Divorced/Widowed	−0.249	-	-
Ongoing psychiatric therapy	-	-	0.959
Being infected with COVID-19	−0.236	-	1.227
Psychiatric therapy modified during COVID-19 pandemic	0.563	1.654	−0.671

PTSD = post-traumatic stress disorder.

## Data Availability

The datasets used in this article are not readily available because individual patient data could not be shared even if de-identified.

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
