# Peer review of "The Economic Impact of Post-Traumatic Stress Disorder Among Patients with Systemic Autoimmune Diseases During the COVID-19 Pandemic"

_ijerph, 2024, doi:10.3390/ijerph21111476_

Round 1
Reviewer 1 Report
Comments and Suggestions for Authors
This is an interesting topic, but the study population is very specific. Can the authors make a case for any sort of generalizability? For example, perhaps they could discuss future disasters and their impact on vulnerable populations? There is also a wide literature on PTSD costs in military veterans. The case needs to be made that these results have public health importance.
Can the authors provide the number of patients excluded from the analytic population? It’s unclear in the current manuscript.
Were participants with previous or non-COVID related PTSD excluded from the study?
The authors might consider adding race/ethnicity to the model unless the sample is homogenous. The authors might also consider adding educational attainment and, if the data are available, some marker of socioeconomic status, such as usual income or occupational level (e.g. service workers vs. professionals).
This sample size is very small. Was a power calculation conducted?
Comments on the Quality of English LanguageThe quallity of the English used in the paper is good but would benefit from minor editing.
Author Response
Comment 1: This is an interesting topic, but the study population is very specific. Can the authors make a case for any sort of generalizability? For example, perhaps they could discuss future disasters and their impact on vulnerable populations? There is also a wide literature on PTSD costs in military veterans. The case needs to be made that these results have public health importance.
Response 1:We thank the reviewer for the comment, indeed we add further comments in the conclusion as follows “The present work may also offer insights into the management of future shocks in such patients, but also serves to reinforce the existing literature on the impact of PTSD following shocks in vulnerable populations.”
Comment 2: Can the authors provide the number of patients excluded from the analytic population? It’s unclear in the current manuscript.
Response 2: We thank the reviewer for the comment, indeed among all patients screened only a few dozen subjects did not complete the assessment and were excluded from the study. As we discussed in the text, the study is part of a bigger project and several efforts were used to ensure high participation, including providing tablet to patients to facilitate them completing all assessment foreseen within the project while waiting for visits, and employment of residents to support patients in completing assessment.
Comment 3: Were participants with previous or non-COVID related PTSD excluded from the study?
Response 3: We thank the reviewer for the comment, that psychiatric comorbidity was uncommon in our study sample. As shown in Table 2, only 19 (6.3%) patients had ongoing psychiatric medication prior to COVID and these were prescribed to address PTSD and other psychiatric comorbidity. Due to the limited number of patients affected, we were not able to further distinguish further patients with psychiatric comorbidity, we take into account that variable in regression models.
Comment 4: The authors might consider adding race/ethnicity to the model unless the sample is homogenous. The authors might also consider adding educational attainment and, if the data are available, some marker of socioeconomic status, such as usual income or occupational level (e.g. service workers vs. professionals).
Response 4: We thank the reviewer for the comment, we didn’t consider race in the study as all the patients were Italian. To what concern marker of socioeconomic status, that we agree may affect results, we considered in model occupational status. We didn’t include education level in models as it was associated with occupational status and using both collinearity problems were found in models. Accordingly we both consider multivariate models including educational level or occupational status as marker of education, after looking at statistics related to model fitting we preferred to use models including occupational status rather than educational level as measure of socioeconomic status.
Comment 5: This sample size is very small. Was a power calculation conducted?
Response 5: We thank the reviewer for the comment, as we elaborated in the text this study is a part of a bigger project. We didn’t have any power analysis for the specific study presented anyway, given the number of predictor and the analyses performed we deemed appropriate the ample size of the study population, despite limited. We discussed about study limitation in the discussion section.
Reviewer 2 Report
Comments and Suggestions for Authors
The article aims to evaluate the economic impact of PTSD in patients with SAD during the COVID-19 pandemic, using a cost-of-illness approach. The authors employ GLMs and LASSO logistic regression to assess the effect of PTSD on different cost components, including healthcare, non-healthcare, and indirect costs. The main conclusion is that while PTSD significantly increases non-healthcare and indirect costs, there is no substantial increase in direct healthcare costs between patients with and without PTSD.
Suggestions to Enhance the Discussion and Address Bias:
The voluntary nature of sample recruitment could lead to selection bias, as patients more willing to participate may have different characteristics from those who opted out, particularly regarding their awareness of PTSD symptoms or their propensity to seek treatment.
It would be reasonable to expect that patients with PTSD incur higher direct healthcare costs due to the complex nature of the condition, which often requires ongoing psychiatric consultations, psychosocial therapies, and medications. The fact that no significant differences in direct healthcare costs were found between patients with and without PTSD raises questions about the adequacy of diagnosis and treatment in this population.
Author Response
The article aims to evaluate the economic impact of PTSD in patients with SAD during the COVID-19 pandemic, using a cost-of-illness approach. The authors employ GLMs and LASSO logistic regression to assess the effect of PTSD on different cost components, including healthcare, non-healthcare, and indirect costs. The main conclusion is that while PTSD significantly increases non-healthcare and indirect costs, there is no substantial increase in direct healthcare costs between patients with and without PTSD.
Suggestions to Enhance the Discussion and Address Bias:
Comment 1: The voluntary nature of sample recruitment could lead to selection bias, as patients more willing to participate may have different characteristics from those who opted out, particularly regarding their awareness of PTSD symptoms or their propensity to seek treatment.
Response 1: We thank the reviewer for the comment, indeed we discussed the fact in the limitation section (see lines 377-380) as we were not able to have the picture of all SAD patients accessing the centre to use statistical methods for addressing possible selection bias.
Comment 2: It would be reasonable to expect that patients with PTSD incur higher direct healthcare costs due to the complex nature of the condition, which often requires ongoing psychiatric consultations, psychosocial therapies, and medications. The fact that no significant differences in direct healthcare costs were found between patients with and without PTSD raises questions about the adequacy of diagnosis and treatment in this population.
Response 2: We thank the reviewer for the comment, indeed the results is a bit surprising and we already provided possible explanations for that finding, in particular the fact that in the pandemic period access to services was generally disrupted and that vulnerable patients also skip visits and treatments more often than in normal condition, or again the social stigma associated with psychiatric disorders may also prevent those patients seeking treatment as they actually need. Moreover results may also be interpreted considering that due to the high direct health costs of SAD patients (because of drugs in particular, but also visits) the impact of costs related to the management of psychiatric comorbidity is relatively low. Another explanation for the finding related to direct costs could be related to the limited follow-up time after diagnosis that prevent looking for additional costs incurred from SAD patients with PTSD. To clarify the issue, in addition to possible explanations already provided in the text, we added a comment in the discussion section lines 356-360.
Reviewer 3 Report
Comments and Suggestions for Authors
1.
Introduction: Avoid short paragraphs. you are encouraged to adhere a more rich structure in introduction. see other published articles for a better insight.
Line 48: You can also discuss other aspects of PTSD including its effects and other similar findings. for example, use this: https://doi.org/10.61838/kman.jppr.2.3.6 and https://journals.kmanpub.com/index.php/aftj/article/view/3195
Materials and Methods:
Line 102: Did you use Italian version? discuss about the used version and its reliability and validity.
Conclusion:
Line 410: This part is weak. you must address the limitations with details. Moreover, the suggestions, both for future studies and implications, must be revised and further discussed.

Author Response
Comment 1:Introduction: Avoid short paragraphs. you are encouraged to adhere a more rich structure in introduction. see other published articles for a better insight.
Response 1:We thank the reviewer for the comment we revised introduction accordingly.
Comment 2: Line 48: You can also discuss other aspects of PTSD including its effects and other similar findings. for example, use this: https://doi.org/10.61838/kman.jppr.2.3.6 and https://journals.kmanpub.com/index.php/aftj/article/view/3195
Response 2: We thank the reviewer for references suggested, we added them in the discussion section rather than in the introduction as we deemed more appropriated in that section.
Comment 3: Materials and Methods:
Line 102: Did you use Italian version? discuss about the used version and its reliability and validity.
Response 3: We thank the reviewer for the comment, we reviewed text specifying that we used the Italian version of both IES and SCI-TALS and when citing we provided a reference about previous study in Italy evaluating validity and reliability of those versions.
Comment 4: Conclusion:
Line 410: This part is weak. you must address the limitations with details. Moreover, the suggestions, both for future studies and implications, must be revised and further discussed.
Response 4: We thank the reviewer for the comment, the sentence was deleted and we end the paragraph saying “The present work may also offer insights into the management of future shocks in similar patients, but also serves to reinforce the existing literature on the impact of PTSD following shocks in vulnerable populations in general”.